# QRS detection in single-lead, telehealth electrocardiogram signals: Benchmarking open-source algorithms

**Florian Kristof**[1], **Maximilian Kapsecker**[1,2], **Leon Nissen**[2], **James Brimicombe**[3], **Martin R. Cowie**[4], **Zixuan Ding**[3], **Andrew Dymond**[3], **Stephan M. Jonas**[2], **Hannah Clair Lindén**[5], **Gregory Y. H. Lip**[6,7], **Kate Williams** [3], **Jonathan Mant**[3], **Peter H. Charlton** [3]*, on behalf of the SAFER Investigators

**1** TUM School of Computation, Information, and Technology, Technical University of Munich, Garching bei München, Germany, **2** Institute for Digital Medicine, University Hospital Bonn, Bonn, Germany, **3** Department of Public Health and Primary Care, University of Cambridge, Cambridge, United Kingdom, **4** School of Cardiovascular Medicine & Sciences, Faculty of Lifesciences & Medicine, King's College London, London, United Kingdom, **5** Zenicor Medical Systems AB, Stockholm, Sweden, **6** Liverpool Centre for Cardiovascular Science at University of Liverpool, Liverpool John Moores University and Liverpool Heart & Chest Hospital, Liverpool, United Kingdom, **7** Danish Center for Health Services Research, Department of Clinical Medicine, Aalborg University, Aalborg, Denmark

* pc657@medschl.cam.ac.uk

**Data Availability Statement:** The following datasets are publicly available at the links provided in Table C in S1 Text: (i) MIT-BIH Arrhythmia

## Abstract

### Background and objectives

A key step in electrocardiogram (ECG) analysis is the detection of QRS complexes, particularly for arrhythmia detection. Telehealth ECGs present a new challenge for automated analysis as they are noisier than traditional clinical ECGs. The aim of this study was to identify the best-performing open-source QRS detector for use with telehealth ECGs.

### Methods

The performance of 18 open-source QRS detectors was assessed on six datasets. These included four datasets of ECGs collected under supervision, and two datasets of telehealth ECGs collected without clinical supervision. The telehealth ECGs, consisting of single-lead ECGs recorded between the hands, included a novel dataset of 479 ECGs collected in the SAFER study of screening for atrial fibrillation (AF). Performance was assessed against manual annotations.

### Results

A total of 12 QRS detectors performed well on ECGs collected under clinical supervision ($F_1$ score $\geq$0.96). However, fewer performed well on telehealth ECGs: five performed well on the TELE ECG Database; six performed well on high-quality SAFER data; and performance was poorer on low-quality SAFER data (three QRS detectors achieved $F_1$ of 0.78-0.84). The presence of AF had little impact on performance.

Database (https://www.physionet.org/physiobank/database/mitdb/); (ii) PhysioNet/Computing in Cardiology Challenge 2014 training dataset and augmented training dataset (https://physionet.org/content/challenge-2014/1.0.0/); (iii) MIT-BIH Normal Sinus Rhythm Database (https://physionet.org/physiobank/database/nsrdb/); and (iv) TELE ECG Database (https://doi.org/10.7910/DVN/QTG0EP). The SAFER dataset cannot be shared due to ethical restrictions. Requests for access to the SAFER dataset should be directed to the SAFER study coordinator (SAFER@medschl.cam.ac.uk) and will be considered by the investigators, in accordance with participant consent.

**Funding:** This study is funded by the British Heart Foundation (FS/20/20/34626 awarded to PHC), and the National Institute for Health and Care Research (NIHR) Programme Grants for Applied Research Programme (RP-PG0217-20007 awarded to JM), and the NIHR School for Primary Care Research (SPCR-2014-10043, project 410 awarded to JM). The funders had no role in study design, data collection and analysis, decision to publish, or preparation of the manuscript.

**Competing interests:** MRC is employed by Astrazeneca PLC. HCL is employed by Zenicor Medical Systems AB.

## Conclusions

The Neurokit and University of New South Wales QRS detectors performed best in this study. These performed sufficiently well on high-quality telehealth ECGs, but not on low-quality ECGs. This demonstrates the need to handle low-quality ECGs appropriately to ensure only ECGs which can be accurately analysed are used for clinical decision making.

## Author summary

The electrocardiogram (ECG) is a vital tool for assessing heart health. Traditionally, ECGs are recorded in clinical settings, but with advances in technology, mobile devices and smartwatches can now be used to record ECGs in daily life. However, ECG recordings from these devices often contain more noise, posing challenges for accurate analysis. In this study, we evaluated 18 different algorithms for detecting heartbeats in ECGs. Our aim was to identify the best-performing algorithm for use with ECGs recorded using mobile devices. We tested each algorithm on 995 ECG recordings and compared their performance against manually-annotated heartbeats. From our analysis, we identified the two best-performing algorithms. These algorithms performed well when analysing high-quality ECGs obtained under clinical supervision and from mobile devices. However, their performance degraded significantly when analysing noisy ECGs from mobile devices. These findings highlight the importance of selecting robust algorithms for ECG analysis, particularly for data collected outside clinical environments. Furthermore, the study demonstrates the need to ensure that only ECGs which can be accurately analysed are used for clinical decision making.

## Introduction

The electrocardiogram (ECG) is one of the most widely used physiological measurement techniques, providing detailed information on heart function. Traditionally ECG measurements have been confined to clinical settings. However, recently it has become possible to measure the ECG in telehealth settings using handheld devices or smartwatches [1, 2]. This presents the opportunity to conduct health assessment beyond the clinical setting, with potential applications including remote health monitoring, personalized diagnosis, rehabilitation, and screening for atrial fibrillation (AF). Indeed, the recent COVID-19 pandemic has acted as a strong catalyst for innovation in this area [3]. However, the increasing use of wearable and telehealth technologies also presents new challenges.

A key challenge is that telehealth ECGs can be of lower quality than those collected in clinical settings, and so ECG analysis algorithms must be able to handle increased noise levels. Telehealth ECGs can be of lower quality for several reasons [4]: the ECG is often measured further away from the heart (such as at the hands rather than the chest); devices typically use dry electrodes rather than the more conductive adhesive electrodes; and there is less quality control since measurements are taken by a non-expert user without clinical supervision. Therefore, there is a need to understand how well ECG analysis algorithms perform in the telehealth setting.

QRS detection is a fundamental task in ECG analysis. QRS complexes indicate ventricular depolarisation, *i.e.* the electrical impulse which causes the the heart to pump blood into the

circulation. QRS detection is widely used for heart rate and rhythm monitoring, and heart rate variability analysis. Furthermore, QRS detection is frequently the first step towards extraction of more detailed ECG features such as QT intervals and P-waves. A range of QRS detection algorithms have been proposed [5, 6], most of which were developed using ECGs collected in clinical settings ([4] being a notable exception). Therefore, there is a need to assess their performance with telehealth ECGs. QRS detectors should firstly be accurate, correctly identifying QRS complexes. They should ideally remain accurate in the presence of pathologies such as AF (which results in an irregular heart rhythm), and in the presence of noise. In addition, QRS detection algorithms should also be stable with low execution times to ensure they are suitable for rapid and long-term analyses.

Previous studies have compared the performance of QRS detection algorithms across databases recorded in different settings. Liu *et al.* assessed ten QRS detectors across five datasets including one telehealth dataset [5]. The algorithms, chosen for their computational efficiency, achieved $F_1$ scores of >99% on high-quality signals, ≤80% for low-quality signals, and ≥94% during pacing and in the presence of arrhythmias. The study concluded that an optimized knowledge-based algorithm [7] performed best. Llamedo and Martinez assessed six QRS detectors on 12 databases covering five categories: normal sinus rhythm, arrhythmia, ST and T morphology changes, stress, and long-term monitoring [6]. The study concluded that the *gqrs* algorithm performed best. Research in [8] assessed 12 QRS detectors across five publicly available datasets. The study concluded that the neurokit (*nk*)algorithm performed best when considering both accuracy and execution time. Previous work in this area addresses known algorithms and benchmark ECG databases, but there is a lack of knowledge about the latest algorithms and their application to new telehealth databases, especially their performance on self-recorded ECGs.

The aim of this study was to identify the best-performing open-source QRS detector for use with telehealth ECGs. The performance of 18 algorithms was assessed on multiple datasets including a novel dataset collected using handheld devices during screening for AF. Performance was assessed primarily in terms of the accuracy of QRS detection (quantified using the $F_1$ score), and also in terms of the execution time and error rate of algorithms. The findings address the gap in knowledge about how well QRS detection algorithms perform in telehealth and settings. They are particularly relevant given the rapid introduction of single-lead ECG technology in consumer devices such as smartwatches, and clinical devices such as handheld ECG recorders.

## Methods

### QRS detection algorithms

The 18 QRS detectors assessed in this study are summarised in Table 1 (with source links provided in Table A in S1 Text). The QRS detectors were identified through a search for opensource algorithms. The majority of algorithms were found in either in the 'NeuroKit' [8] or 'ecgdetector' [9] Python packages. Some algorithms were available in both packages with slightly different implementations, in which case the faster implementation was used. Python implementations were used where available to provide a fair comparison of algorithm execution times. In four cases Python implementations were not available and Matlab implementations were used instead (*jqrs*, *rdeco*, *rpeak*, and *unsw*). Six additional algorithms were identified but not used in this study due to one of the following reasons: (i) no Python or Matlab implementation was available; (ii) the available implementation only accepted particular sampling frequencies; (iii) the available implementation predominantly led to errors; or

**Table 1. Datasets.**

| Description | No. Beats | No. Recordings | Recording Duration (min) | Total Time (min) | Sampling Frequency (Hz) | Source |
|---|---|---|---|---|---|---|
| *Supervised ECG recordings* | | | | | | |
| **SIN:** Recordings from patients without arrhythmias: excerpts from long-term recordings collected at an Arrhythmia Laboratory. | 185,253 | 18 | 120 | 2,160 | 500 | MIT-BIH NSR database [10] |
| **ARR:** Recordings from patients with and without arrhythmias: excerpts from 24-hour ambulatory recordings collected at an Arrhythmia Laboratory. | 112,599 | 48 | 30 | 1,440 | 360 | MIT-BIH arrhythmia database [10, 11] |
| **HIGH:** High-quality recordings from patients and healthy volunteers: collected from multimodal devices such as bedside monitors. | 72,315 | 100 | 10 | 1,000 | 250 | 2014 PhysioNet/CinC challenge training set [10, 12] |
| **LOW:** Low-quality recordings from patients and healthy volunteers: collected from multimodal devices such as bedside monitors. | 78,518 | 100 | 10 | 1,000 | 360 | 2014 PhysioNet/CinC challenge augmented training set [10, 12] |
| *Unsupervised, telehealth ECG recordings* | | | | | | |
| **TELE:** Single-lead, telehealth ECGs from home-dwelling patients: collected by patients without supervision using a device which records the ECG from the hands. | 5,932 | 250 | 0.50 | 125 | 500 | Harvard dataverse TELE database [4] |
| **SAFER:** Single-lead, telehealth ECGs from home-dwelling AF screening participants: collected by participants without supervision using a handheld device. Split into subsets according to presence of AF (AF or non-AF) and ECG quality (HIGH or LOW): | 18,279 | 479 | 0.50 | 239.5 | 500 | SAFER Feasibility Study (private) [13] |
| - SAFER-AF-HIGH | 8,456 | 183 | 0.50 | 91.5 | 500 | |
| - SAFER-nonAF-HIGH | 7,065 | 199 | 0.50 | 99.5 | 500 | |
| - SAFER-nonAF-LOW | 2,758 | 97 | 0.50 | 48.5 | 500 | |

(iv) the execution time was substantially longer than that of other algorithms. Further details are provided in Table B in S1 Text.

## Datasets

The performance of QRS detectors was assessed using six datasets, including datasets collected in inpatient, outpatient, and home settings. The datasets are summarised in Table 2 and described in the following paragraphs. Full source links for the datasets are provided in Table C in S1 Text.

**MIT-BIH Normal Sinus Rhythm Database (SIN).**   The MIT-BIH Normal Sinus Rhythm Database (SIN) contains 18 24-hour ECG recordings from patients referred to the Arrhythmia Laboratory at Boston's Beth Israel Hospital, who were found not to have significant arrhythmias [10]. The first two hours of each recording were used in this study. The subjects consisted of 13 women and 5 men, aged 20 to 50. Each recordings contains two ECG channels of unknown leads, the first of which was used in this analysis.

**MIT-BIH Arrhythmia Database (ARR).**   The MIT-BIH Arrhythmia Database (ARR) contains 48 30-minute ECG recordings from 47 patients referred to the same Arrhythmia Laboratory [10, 11]. This dataset consists of 23 recordings which were selected at random from a larger dataset and a further 25 recordings which were manually selected to include examples of significant but uncommon arrhythmias. The subjects included 22 women and 25 men aged 23 to 89. The first ECG channel in each recording was analysed, which was the modified limb lead II in most cases.

**Table 2. QRS detection algorithms.**

| Abbreviation | Description | Reference(s) |
|---|---|---|
| christ | **christov:** Detect QRS complexes as points exceeding an adaptive threshold consisting of the sum of: (i) steep-slope threshold (a linear reduction from 200ms to 1200ms 200ms after a QRS); (ii) adaptive integrating threshold (increases in the presence of electromyogram noise); and (iii) adaptive beat expectation threshold (a linear reduction between 2/3 and 1 mean RR-interval after a QRS). | [14] |
| engz | **engzee, sqrs:** Detection of QRS complexes as points exceeding a threshold based on a filtered signal: (i) removal of baseline wandering; (ii) application of a threshold to detect R-peaks; (iii) application of a refractory period to prevent multiple detection of a single R-peak. | [15, 16] |
| fnvg | **FastNVG:** A natural visibility graph (NVG) based R-peak detector: (i) representing the ECG signal as a graph using NVG; (ii) calculation of a node metric in the graph domain for weighting the signal to emphasize R-peak positions (iii) applying thresholding as with pan-tomp. | [17, 18] |
| fwhvg | **FastWHVG:** An R-peak detector based on the horizontal visibility graph (HVG): The method is similar to the fnvg algorithm, but is computationally more efficient since the HVG is a subset of the NVG. | [17, 18] |
| gamb | **gamboa:** Detection using amplitude histogram and critical points: (i) signal normalization using the amplitude histogram; (ii) detection of critical points in the first derivative exceeding a threshold; (iii) elimination of false beats through constraints on detected ECG signal beats; (iv) Computation of the mean ECG wave to obtain QPRS features. | [19] |
| gqrs | **gqrs:** The ECG beat detection algorithm initiates with: (i) employing a trapezoid low-pass filter to the signal, followed by a QRS matched filter convolution. (ii) The parameters of recent intervals and peak thresholds are adjusted without recording QRS locations. (iii) Sample detection occurs, identifying larger samples and peaks that surpass the QRS threshold, thus marking them as QRS complexes. If no peak is detected, the system lowers the peak detection threshold. (iv) Primary and secondary peak identification differentiates between peak types based on neighborhood size and relevance to a previous primary peak or associated T-wave. | [10] |
| hamilt | **hamilton, eplimited:** Detect QRS complexes using filtering, differentiation, rectification, and a moving window method: (i) applying low-pass and high-pass filtering to the signal; (ii) calculating the signal's derivative; (iii) rectifying the signal and utilizing a moving window of 80 ms; (iv) detecting QRS complexes following a predefined rule set. | [20, 21] |
| jqrs | **jqrs:** QRS detection enhanced by sliding window and custom filter: (i) window-based peak energy detector; (ii) band-pass filter with QRS matched filter (Mexican hat); (iii) reject detections based on heuristic during flat lines; (iv) search-back procedure for suspected missed beats. | [22–24] |
| kali | **Kalidas and Tamil, Stationary Wavelet Transform (swt):** Peak detection using Stationary Wavelet Transform: (i) resample signal to 80 Hz for real-time processing; (ii) compute 2-level SWT using 'db3' wavelet; (iii) square and MWA to enhance QRS peaks; (iv) threshold-based peak detection; (v) detect missed beats based on RR intervals; (vi) determine actual R-peak location within 0.10 seconds. | [8, 25] |
| mart | **martinez, wavedet, Continuous Wavelet Transorm (CWT):** The algorithm functions by executing a continuous wavelet transformation of the ECG signal across five distinct scales: (i) Each scale calculates a standard deviation, epsilon, from the transformed signal and peaks exceeding this epsilon are identified. (ii) The algorithm then filters these identified peaks across each scale, keeping only those closely associated with preceding scale peaks. (iii) It locates R-peaks by pinpointing zero-crossings in scale one within a specified range. | [26] |
| nab | **nabian:** Usage of sliding window with adaptive thresholds and domain knowledge to detect PQRST points: (i) Filter using Elliptic, Gaussian, or Butterworth (default: Elliptic); (ii) Detect potential R-peaks using global maxima in sliding window; (iii) Eliminate R-peaks below amplitude threshold; (iv) Find missing R-peaks using R-R interval; (v) Detect PQST points using predefined R-based locations. | [27] |

(*Continued*)

**Table 2.** (Continued)

| Abbreviation | Description | Reference (s) |
|---|---|---|
| nk | **neurokit:** Usage of signal smoothing and gradients to detect QRS complexes: (i) Computing the gradient and average gradient threshold of the highpass-filtered raw ECG signal. (ii) Identifying the start and end of QRS complexes by comparing the signal's smoothed gradient with the gradient threshold. (iii) Ignoring QRS complexes that are too short by setting a minimum length. (iv) Identifying R-peaks within each QRS. (v) Ensuring peaks identified are not too close together by enforcing a minimum delay. | |
| pan-tomp | **pan tompkins:** Filtering of the signal to segment the QRS complex: (i) low-pass and high-pass filtering; (ii) derivative of the signal; (iii) squaring of the derivative to amplify the QRS complex; (iv) adaptive thresholding with a refractory period using a moving window approach. | [28] |
| rdeco | **r-deco:** QRS detection using an envelope-based method: (i) using the difference between the lower and upper envelopes to flatten the signal; (ii) limiting the search range by considering segments whose value is higher than the 80 ms later value and whose upward slope lasts longer than 80 ms; (iii) selecting the segments of maximal value; (iii) defining the R peaks using the Pan-Tompkin adaptive thresholding method; (iv) eliminating false detections by performing a 50 ms backward search for each peak. | [29] |
| rpeak | **rpeakdetect:** Periodic adjustment of thresholds and parameters to detect QRS complexes using sensitivity-appropriate filtering: (i) cascaded low-pass and high-pass filtering to reduce signal noise; (ii) approximation of a derivative and application of an amplitude squaring operation; (iii) use of a moving window integrator with adaptive thresholds to segment the locations of QRS complexes. | [21, 28] |
| two-avg | **two average, elgendi:** Application of statistical thresholds and moving averages to generate blocks of interest: (i) bandpass filtering; (ii) integration of moving averages to generate blocks of interest; (iii) rejection of blocks smaller than the QRS complex length of the healthy adult and detection of the R peak as the maximum value of the remaining blocks. | [30] |
| unsw | **unsw:** Application of an adaptive threshold to a feature enriched signal: (i) filtering the signal with detrending, median filtering, and bandpass filters; (ii) calculating the QRS feature using the differentiated and filtered signal; (iii) smoothing the QRS features signal frequencies using the fundamental frequency as the lower bound; (iv) Adaptive threshold calculation using windows of different lengths on the filtered QRS feature signal; (v) Identification of possible QRS complexes using a peak-through detector on the filtered QRS feature signal; (vi) Rejection of erroneous QRS complexes. | [4] |
| wqrs | **wqrs:** Transformation to a curve-length signal to apply an adaptive threshold: (i) low-pass filtering; (ii) non-linear scaling of the signal to amplify the QRS complex and reduce noise; (iii) an adaptive threshold reveals the onset and duration of the QRS complex. | [31] |

**PhysioNet/Computing in Cardiology Challenge 2014 Datasets (HIGH and LOW).** The PhysioNet/Computing in Cardiology Challenge 2014 datasets consist of 10-minute ECG recordings from patients and healthy volunteers [10, 12]. The two publicly available datasets were used in this study: (i) the Training Set (HIGH), which contains 100 recordings which are generally of high quality; and (ii) the Augmented Training Set (LOW), which contains 100 recordings that are generally of low quality. Each record in these datasets contains a single ECG lead. The LOW dataset contains the following leads: lead II (78 records); lead III (5), lead AVF (3), lead AVL (1), and no lead label (13). No lead labels are provided in the HIGH dataset.

**TELE ECG database (TELE).** The TELE ECG Database contains 250 30-second lead-I ECG recordings from home-dwelling patients suffering from chronic obstructive pulmonary disease and/or congestive heart failure [4, 32]. Recordings were acquired without clinical

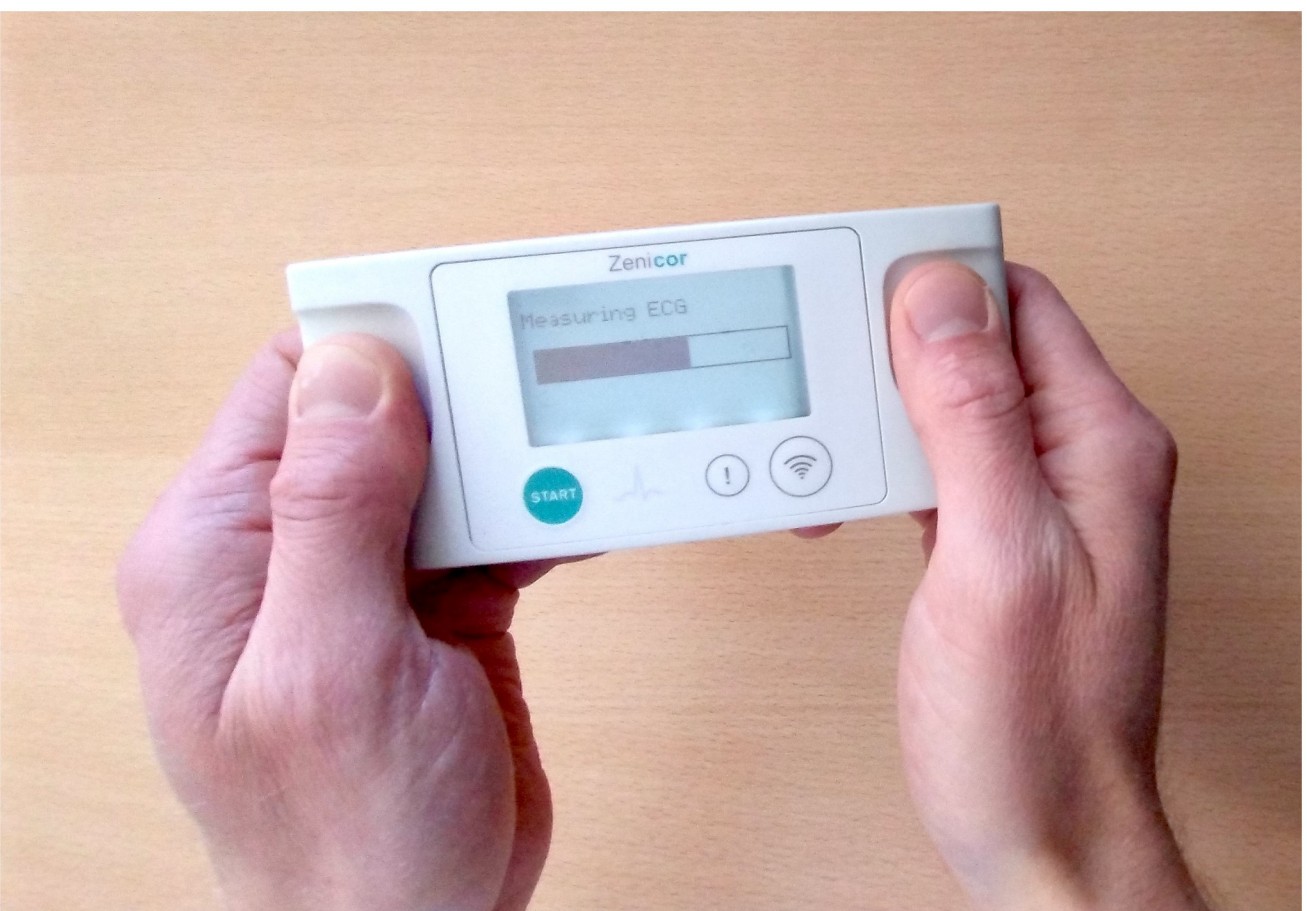

**Fig 1. Zenicor-EKG device.** The handheld Zenicor-EKG device used to record 30-second ECGs in the SAFER ECG Dataset.

supervision using the TeleMedCare Health Monitor (TeleMedCare Pty. Ltd. Sydney, Australia). The device records an ECG from the hands using dry metal electrodes. This dataset contains 221 ECGs randomly selected from 120 patients, and an additional 29 ECGs specifically selected to represent poor-quality data. The dataset contains manual annotations of QRS complexes. One ECG in the dataset lasted longer than 30s, and was truncated to 30s for this study.

**SAFER ECG dataset (SAFER).**   The SAFER ECG Dataset contains 479 30-second lead-I ECG recordings from home-dwelling subjects aged 65 and over, collected in an AF screening study (the SAFER Feasibility Study, ISRCTN 16939438) [13].

ECG recordings were acquired without clinical supervision using the Zenicor EKG-2 device shown in Fig 1 (Zenicor Medical Systems AB, Sweden). The device records an ECG from the thumbs using dry metal electrodes. This dataset contains: 183 high-quality ECGs exhibiting AF (denoted SAFER-AF-HIGH) collected from 48 subjects (13 female and 35 male); 199 high-quality ECGs from subjects without AF (SAFER-nonAF-HIGH) collected from 199 participants (100 female and 99 male); and 97 low-quality ECGs from subjects without AF (SAFER-nonAF-LOW) collected from 97 subjects (49 female and 48 male). ECG quality was assessed using the Cardiolund ECG Parser algorithm (Cardiolund AB). R-peaks were manually annotated specifically for this study. The presence of AF was determined as described in [13]: (i) using the Cardiolund algorithm to identify ECGs with potential abnormalities; and (ii) expert

reviewers manually reviewing ECGs to identify AF (as described in [13, 33]). To provide further details, ECGs were classified as AF and non-AF based on *ad hoc* review by two cardiologists. An ECG was classified as AF if either: (i) both cardiologists agreed that the ECG contained AF; or (ii) one cardiologist made an AF diagnosis and the other provided no diagnosis. An ECG was classified as non-AF if either: (i) the Cardiolund algorithm didn't identify abnormalities in the ECG, and the cardiologists did not identify an arrhythmia, and the participant was not diagnosed with AF; or (ii) both cardiologists agreed that the ECG didn't contain an arrhythmia.

## Ethics statement

The SAFER Feasibility Study in which the SAFER ECG dataset was acquired was approved by the London Central NHS Research Ethics Committee (18/LO/2066). All participants gave written informed consent to participate in the study. The study was conducted in accordance with the Declaration of Helsinki. Ethical approval was not required for the use of the remaining datasets as these were pre-existing, anonymised datasets.

## Statistical analysis

The performance of QRS detectors was primarily assessed using the $F_1$ score (following a precedent in [5, 34]). The $F_1$ score is the harmonic mean of the sensitivity (SEN) and positive predictive value (PPV). These three statistics were calculated as follows from: the number of reference QRS complex annotations ($n_{ref}$, corresponding to the number of actual positives, P); the number of QRS complexes identified by an algorithm ($n_{alg}$, corresponding to the number of predicted positives, *i.e.* true positives + false positives, TP+FP); and the number of QRS complexes which were correctly identified ($n_{correct}$, corresponding to the number of true positives, TP).

$$SEN(\%) = \frac{TP}{P} \times 100 = \frac{n_{correct}}{n_{ref}} \times 100 \tag{1}$$

$$PPV(\%) = \frac{TP}{TP + FP} \times 100 = \frac{n_{correct}}{n_{alg}} \times 100 \tag{2}$$

$$F_1(\%) = \frac{2 \times PPV \times SEN}{PPV + SEN} \times 100 \tag{3}$$

$n_{correct}$ was calculated as the number of reference QRS complex annotations for which at least one QRS complex was identified by an algorithm within ± 75ms of the reference QRS annotation as shown in Fig 2.

A threshold of ± 75ms was chosen to classify QRS detections as correct or not for the following reasons. QRS complexes typically last <120ms in health [35], although can last longer in disease [36]. ± 75ms was identified as a conservative threshold which would classify any QRS detections lying on a QRS complex as correct, whilst classifying any detections on other ECG waves (such as p- or t-waves) as incorrect. This was based on the assumptions that: a QRS complex lasts up to approximately 150ms; the R-wave is located approximately in the centre of a QRS complex; and reference QRS annotations are at the locations of R-waves. To investigate the suitability of this threshold, we assessed the performance of the QRS detectors on the telehealth (TELE and SAFER) datasets for thresholds ranging from ± 1 to 140ms. The results (shown in Fig A in S1 Text) show that for all QRS detectors performance was poorer at low thresholds, with performance generally approaching a maximum between 20 and 100ms

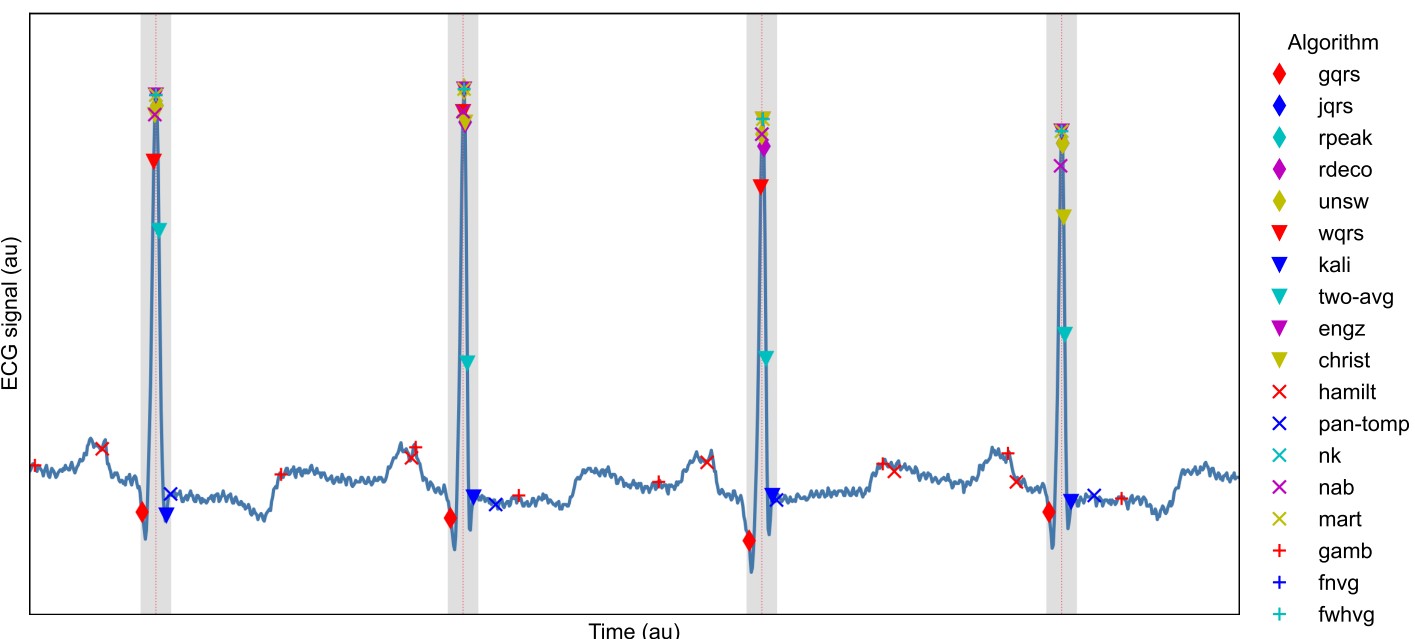

**Fig 2. Assessing whether QRS complexes were correctly identified.** An ECG signal is shown with dotted red lines marking reference R-peak annotations, grey areas showing the tolerance of ± 75ms around these annotations within which QRS complexes are deemed to be correctly identified, and markers for the R-peaks identified by the 18 QRS detectors used in this study.

(such as $\approx$ 20ms for *nk* and *unsw*, and $\approx$ 60–80ms for *pan-tomp* and *two-avg*). Therefore, a threshold of ± 75ms appeared a reasonable choice. In comparison, previous work in this area has used tolerances of 50 ms [5] and ± 150 ms [22].

$F_1$ scores are reported using the median and inter-quartile range of the $F_1$ score for each ECG window.

Two additional performance measures were used: algorithm error rate and execution time. Error rates were defined as the percentage of 30s ECG segments in which an algorithm encountered an error and did not return identified QRS complexes. Execution times were assessed as the median time taken for an algorithm to process each 30s ECG segment. The analysis was performed on a MacBook Air (M1, 2020, 16 GB RAM, 8 cores) without parallelization. The assessment was run in Visual Studio Code 1.73.0, using Python 3.9, and calling MATLAB R2022a for QRS detectors written in MATLAB code.

The two-sided Mann-Whitney U test was used to test for statistically significant differences between $F_1$ scores at the 95% significance level. A Bonferroni correction was used to account for the multiple comparisons (a comparison for each beat detector). This test was used as the distributions were neither normally distributed nor dependent on each other. Comparisons were made between: (i) supervised and telehealth ECGs; (ii) high- and low-quality ECGs; (iii) AF and non-AF ECGs; and (iv) female and male subjects. Comparisons between female and male subjects were made on the SIN, ARR and SAFER datasets, but not on the HIGH, LOW and TELE datasets as to the best of our knowledge they do not contain information on gender.

## Results

### Algorithm performance

The performance of the algorithms is presented in Fig 3 using the $F_1$ score. When using a $F_1$ score of $\geq$ 0.96 to identify good performance, a total of 12 out of 18 algorithms performed well

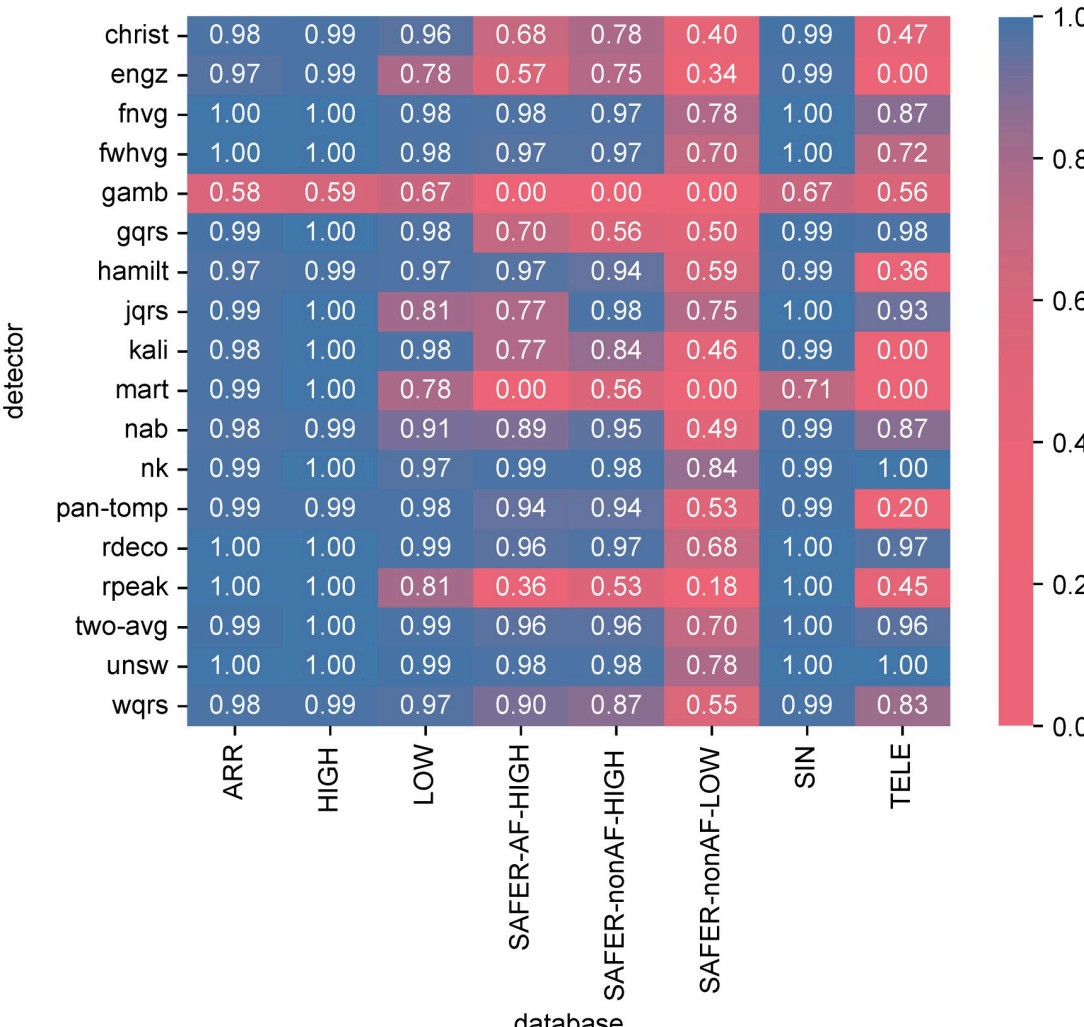

**Fig 3. The performance of QRS detectors, expressed as the $F_1$ score.** Results are shown for the 18 QRS detectors (on the y-axis) and the six datasets (x-axis). *Dataset definitions: ARR*—MIT-BIH Arrhythmia Database; *HIGH*—PhysioNet/Computing in Cardiology Challenge 2014 training set; *LOW*—PhysioNet/Computing in Cardiology Challenge 2014 augmented training set; *SAFER-AF-HIGH*—SAFER ECG Dataset subset of high-quality ECGs exhibiting AF; *SAFER-nonAF-HIGH*—SAFER ECG Dataset subset of high-quality ECGs not exhibiting AF; *SAFER-nonAF-LOW*—SAFER ECG Dataset subset of low-quality ECGs not exhibiting AF; *SIN*—MIT-BIH Normal Sinus Rhythm Database; *TELE*—TELE ECG Database.

on ECGs collected under clinical supervision (ARR, HIGH and LOW, and SIN). The exceptions were *engz, gamb, jqrs, mart, nab* and *rpeak*. Fewer algorithms performed well on telehealth ECGs: five algorithms performed well on the TELE dataset (*gqrs, nk, rdeco, two-avg,* and *unsw*); six algorithms performed well on high-quality SAFER data (*fnvg, fwhvg, nk, rdeco, two-avg,* and *unsw*); and performance was considerably poorer on low-quality SAFER data, with only three algorithms scoring $\geq$ 0.78 (*fnvg, nk,* and *unsw*), and none scored higher than 0.84.

Therefore, overall the *nk* and *unsw* algorithms performed best, with consistently high $F_1$ scores on datasets of supervised ECG recordings, and the highest $F_1$ scores on self-recorded ECGs (TELE and SAFER datasets).

Additional results for the positive positive predictive value (PPV) and sensitivity (SEN) are provided in Figs B and C in S1 Text. These metrics show that: *gamb* performed poorly because

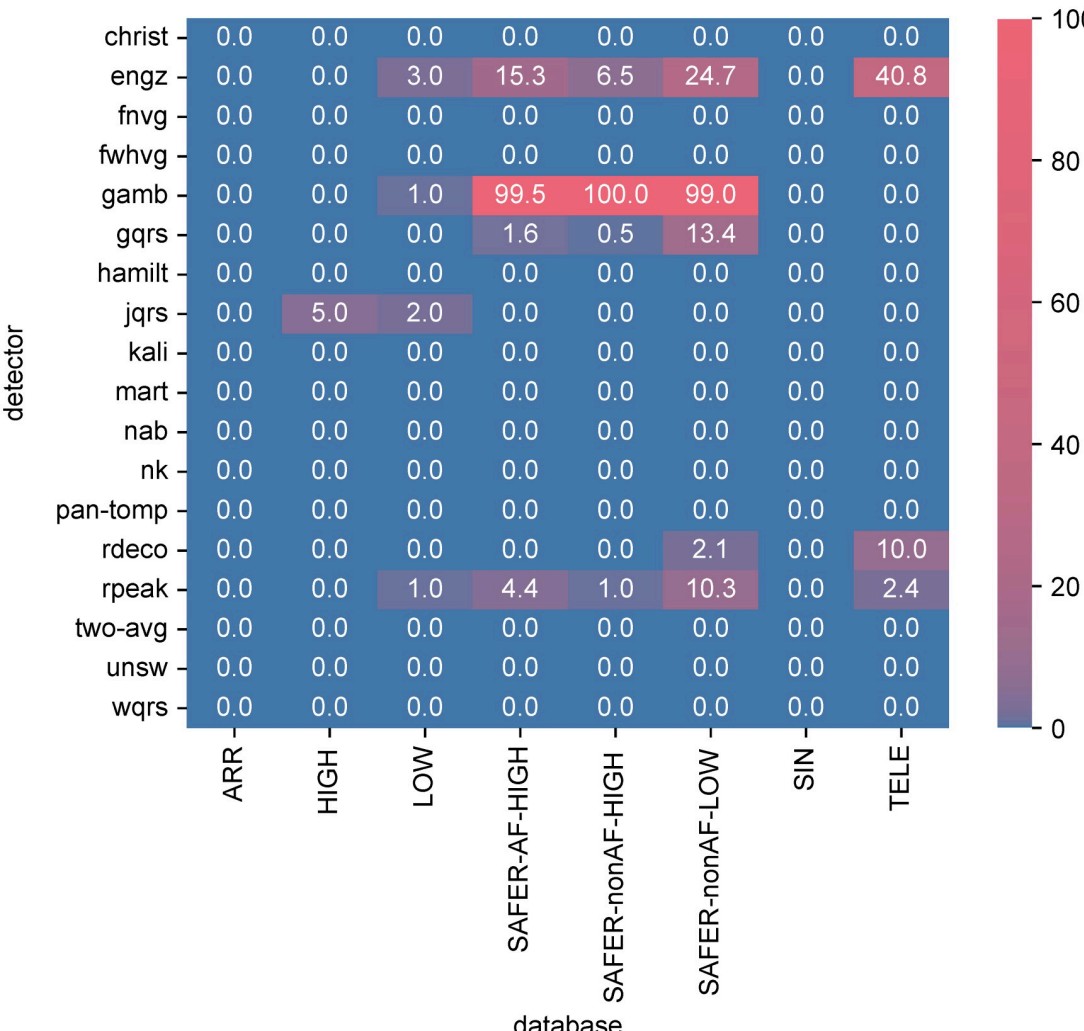

**Fig 4. The error rates for each QRS detector (expressed as percentages).** These indicate the proportion of ECG windows for which the QRS detector algorithms failed to execute (*i.e.* encountered an error).

of a low PPV, indicating that it falsely detected additional QRS complexes; and *mart*, and *engz* had a low SEN, indicating that they frequently missed QRS complexes.

Fig 4 shows the error rates of each QRS detector (indicating the proportion of ECG windows for which the QRS detector algorithms failed to execute—*i.e.* encountered an error). Most QRS detectors had no or very few errors. The best-performing algorithms had 0.0% errors on all datasets (*nk* and *unsw*). The *engz* and *gamb* algorithm implementations frequently produced errors, and some errors were encountered for *gqrs*, *jqrs*, *rdeco* and *rpeak* algorithms. Of particular note, the *gamb* algorithm exhibited higher error rates on SAFER data, including error rates of ≥99% for *gamb* (in keeping with a previous study [8]). This was due to the algorithm's use of a fixed amplitude threshold which was often not met for SAFER ECGs.

Fig 5 shows the median execution time of each QRS detector. The fastest QRS detector, *rpeak*, had an execution time of 1.1 ms (i.e. 0.004% of the signal duration). Of the best-performing QRS detectors (*nk* and *unsw*), *nk* had a short execution time of 2.7 ms (0.009% of the signal duration), whereas *unsw* was slower at 37.1 ms (0.124% of the signal duration). Four

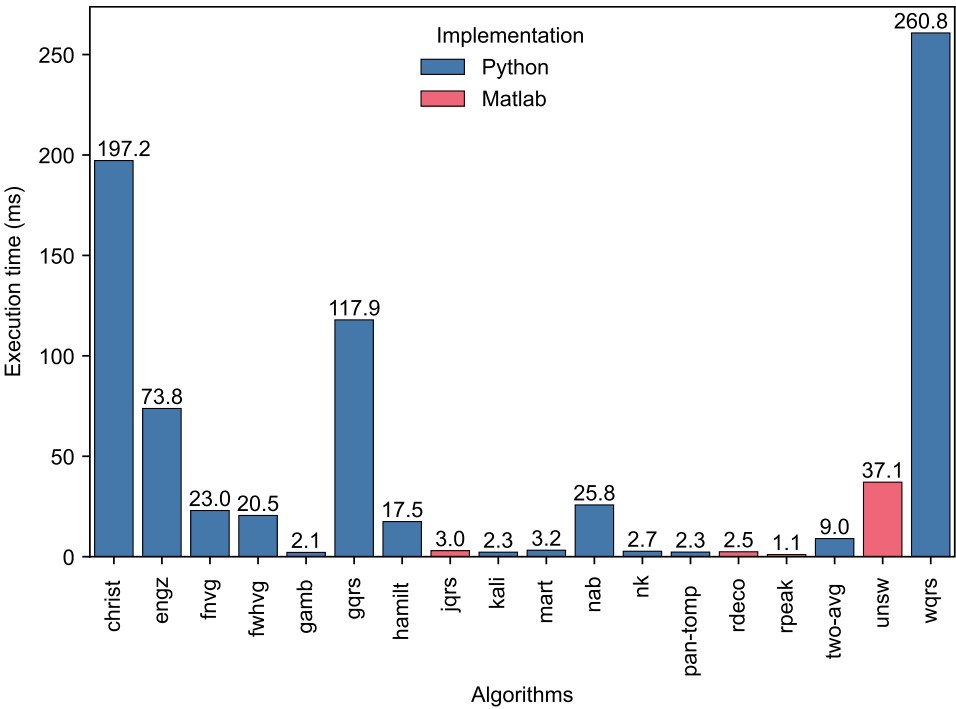

**Fig 5. QRS detector execution times.** The median execution time of each QRS detector was calculated across all datasets, where QRS detectors were implemented in either Python (blue) or Matlab (red).

QRS detectors had much longer execution times (*christ*, *engz*, *gqrs*, and *wqrs*), although we note that C code implementations are available for some of these that would have led to shorter execution times. Most QRS detectors had similar median and mean execution times: the mean execution time was between 87 and 124% of the median for all QRS detectors except *christ*, whose median execution time was substantially longer (373% of the median), primarily due to exceptionally high runtimes on the SAFER-nonAF-LOW dataset.

## Comparison between supervised and telehealth ECGs

For most QRS detectors, the performance of QRS detectors was higher on supervised ECG recordings than on unsupervised, telehealth ECGs. A total of 17 (out of 18) QRS detectors had a significantly higher $F_1$ score on the supervised SIN dataset than the unsupervised SAFER-nonAF-HIGH dataset (*mart* showed no sigificant difference). Similarly, 16 QRS detectors had a significantly higher $F_1$ score on the supervised ARR dataset than the unsupervised SAFE-R-AF-HIGH dataset (*hamilt* and *nk* showed no sigificant difference). Referring to Fig 3: some QRS detectors performed below average on unsupervised ECGs despite having performed well on supervised ECGs: *gqrs* achieved $F_1$ scores of ≥0.98 on supervised ECGs (SIN, ARR, HIGH, LOW), but ≤0.70 on SAFER; and *rpeak* achieved ≥0.81 on supervised ECGs, but ≤0.53 on TELE and SAFER datasets.

The results for positive predictive value (PPV) and sensitivity (SEN) (in Figs B and C in S1 Text) show that most QRS detectors which performed poorly on self-recorded ECGs had a low PPV, indicating false positive QRS detections. In addition, some QRS detectors had low sensivities, indicating unrecognized QRS complexes (e.g. *engz*, *gamb*, *jqrs*, *mart*, *nab*, and *wqrs*).

Algorithm errors predominantly occurred in unsupervised telehealth ECGs (see Fig 4).

### The impact of signal quality

Low signal quality was associated with poorer performance of QRS detectors in the telehealth setting. The $F_1$ scores for all QRS detectors except *gamb* were significantly lower on low-quality unsupervised ECGs (SAFER-nonAF-LOW) than high-quality unsupervised ECGs (SAFER-nonAF-HIGH). For instance, the best-performing QRS detectors (*nk* and *unsw*) performed well on high-quality unsupervised ECGs (TELE, SAFER-nonAF-HIGH, SAFER-AF-HIGH) with $F_1$ scores of ≥0.97, but performed less well on low-quality unsupervised ECGs (SAFER-nonAF-LOW) with $F_1$ scores of ≤0.84. Indeed, all remaining QRS detectors showed $F_1$ scores of ≤0.78 on low-quality ECGs (SAFER-nonAF-LOW) in the unsupervised telehealth environment.

Signal quality had a smaller but nonetheless significant impact on QRS detectors when using supervised ECGs. Almost all algorithms performed well on high-quality ECGs (the SIN, ARR, and HIGH datasets) with $F_1$ scores of ≥0.97 (except *gamb* and *mart*), and most of these algorithms continued to perform relatively well on low-quality supervised ECGs (the LOW dataset) with $F_1$ scores of ≥0.97 (except *engz*, *gamb*, *jqrs*, *mart*, *nab* and *rpeak*). The small differences in $F_1$ scores between HIGH and LOW were significant for all QRS detectors except *wqrs* and *hamilt*.

### Other influencing factors

The presence of arrhythmia did not have a large effect on $F_1$ scores for either supervised ECGs (comparing ARR and SIN) or unsupervised ECGs (comparing SAFER-AF-HIGH and SAFER-nonAF-HIGH) (see Fig 3). Whilst the differences were mostly small, $F_1$ scores were significantly lower during arrhythmias in ARR compared to SIN for 6 out of 18 QRS detectors, and in SAFER-AF-HIGH compared to SAFER-nonAF-HIGH for 8 QRS detectors. Amongst the best performing QRS detectors (*nk* and *unsw*), the only significant difference was for *nk* in the comparison of SAFER-AF-HIGH and SAFER-nonAF-HIGH, although this difference was small with median $F_1$ scores of 0.99 and 0.98 on the datasets.

Sex had little impact on performance when using unsupervised ECGs as demonstrated by there being no significant differences in performance between female and male subjects on high-quality, non-AF SAFER signals (see Fig 6A), and significant differences for only two QRS detectors on high-quality, AF SAFER signals (see Fig 6B). There were no significant differences in performance between sexes on the SIN and ARR datasets (see Fig D in S1 Text), although we note the small numbers of subjects in each group in the SIN dataset (13 female and 5 male).

p-values for all statistical comparisons are provided in Tables D and E in S1 Text.

## Discussion

### Summary of findings

This study assessed the performance of open-source QRS detectors on single-lead, telehealth ECGs. The neurokit (*nk*)and UNSW (*unsw*)QRS detectors were identified as the best-performing out of 18 QRS detectors. They performed well on telehealth ECGs recorded without clinical supervision, and also on ECGs recorded in clinical settings. They achieved $F_1$ scores of ≥0.98 on high-quality telehealth ECGs and ≥0.97 on ECGs recorded in clinical settings. Performance was lower at ≥0.78 when analysing low-quality telehealth ECGs. Performance was not substantially affected by heart rhythm or gender. *nk* had one of the fastest execution times (at 0.009% of the signal duration), whereas *unsw* was over ten times slower (0.124%).

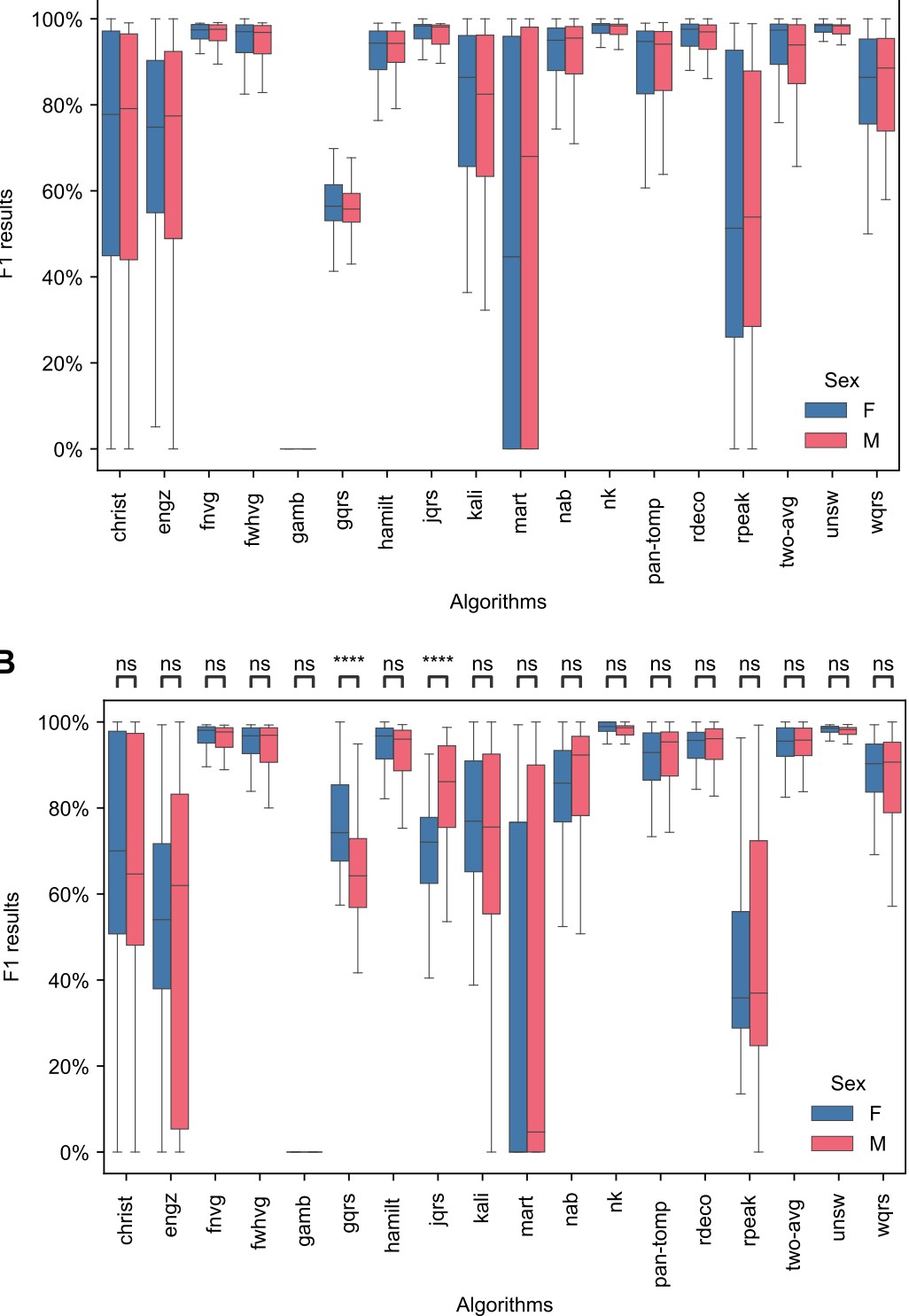

**Fig 6. Comparison of the performance of QRS detectors between female (F) and male (M) SAFER participants.** A: SAFER-nonAF-HIGH: High-quality, non-AF ECGs (including 100 female and 99 male subjects). B: SAFER-AF-HIGH: High-quality, AF ECGs (including 92 female and 91 male subjects). *Definitions: ns—no significant difference.*

## Comparison with literature

Several studies have compared the performance of multiple QRS detection algorithms across databases of different quality [5, 6, 8, 22]. Previous studies assessed 6–12 algorithms, compared to 18 in the current study. Several of the high-performing algorithms included in the current study were not widely assessed in previous comparison studies: *nk* and *two-avg* were only included in [8]; *unsw* was only included in [5]; and *rdeco* was not included in these studies. In addition, previous studies had mostly focused on assessing performance on supervised ECG recordings rather than the telehealth setting. Telehealth data was only included in [5]: the current study included analyses of both this dataset and also data from the SAFER AF screening study, containing the additional challenge of QRS detection during AF.

The current study adds to our understanding of how best to detect QRS complexes in telehealth ECGs, and demonstrates the need to develop techniques to handle low-quality ECGs appropriately. Previously, QRS detectors had been found to perform worse on telehealth data, and in particular the TELE dataset [5]. We also observed worse performance on telehealth data, although we found that the best QRS detectors performed adequately well on high-quality telehealth data, and that performance was only substantially worse on low-quality telehealth data. This provides two complementary directions for future work: (i) QRS detectors could be developed to perform well even in the presence of noise (*e.g.* through denoising [37] or improved algorithm design [4]); and (ii) ECG signal quality algorithms could be developed to identify low-quality recordings in which QRS complexes cannot be accurately identified [22, 38].

The current study also has implications for future research. We observed that the performance of QRS detectors on supervised or high-quality ECG recordings is not necessarily indicative of their performance on unsupervised recordings, in keeping with [6]. This highlights the importance of assessing performance in the target setting, such as in AF screening as performed in this study. We also observed quite different performances on the TELE dataset to those reported previously: whereas the highest performing algorithm achieved an $F_1$ score of 0.80 on TELE in [5], six of the algorithms included in the present study achieved $F_1$ scores of 0.90–1.00. Whilst in many cases this may be explained by including additional algorithms in this study, it is notable that the *jqrs* algorithm's performance was substantially higher on this dataset in the present study (0.93) than the previous study (0.79). This may also be explained by the use of different tolerance windows. Nonetheless, this demonstrates the need to share open-source algorithm implementations and the code used to perform algorithm assessments. To address this, we have provided a repository of open-source algorithms and assessment code to accompany this article: https://github.com/floriankri/ecg_detector_assessment.

## Strengths and limitations

The key strengths of this study are the assessment of QRS detectors in a real-world AF screening setting, and the inclusion of recently developed, high-performance QRS detectors. The key limitation is that algorithms were run retrospectively on a computer, rather than in real-time on a telehealth device. Some algorithms were implemented in Python, and others in Matlab. Therefore, the comparison of algorithm execution times reported in this study may not be truly representative of the relative execution times which would be observed on devices: the comparison of Python and Matlab execution times may not be fair; different algorithms may have been optimised to different extents; and some algorithms may be more amenable to further optimisation for use on devices than others (such as through implementation in C, as is already the case for parts of *unsw*). We note that in this study we did not investigate the potential benefit of additional ECG filtering beyond that already incorporated into each of the QRS

detector algorithms: potentially performance could be improved further by including additional linear or non-linear filtering steps [39, 40]. Furthermore, we did not investigate the accuracy of RR-intervals derived from QRS detections, nor their suitability for heart rate variability analysis or arrhythmia detection. We note that additional processing steps may be required to accurately derive RR-intervals, such as locating the R-wave on each detected QRS complex.

### Implications

This study identified leading QRS detector algorithms for use with telehealth ECGs. The best-performing algorithms were able to detect QRS complexes with a very high degree of accuracy on high-quality telehealth ECG data, demonstrating the potential utility of telehealth devices for assessments based on RR-intervals (such as arrhythmia detection). Furthermore, the study demonstrates the importance of selecting a high-performance QRS detector, since performance can vary greatly on telehealth ECGs, between even well-established algorithms. The study also demonstrates the difficulty in analysing low-quality telehealth ECGs, which appear to be of particularly low quality, perhaps due to increased artifact, the use of dry electrodes, being self-recorded without clinical supervision, and acquisition at the hands rather than the chest [4].

The findings are particularly relevant to telehealth settings where ECG signals are recorded without clinical supervision. Several such settings arise in the detection and management of atrial fibrillation at home, including: (i) virtual wards to reduce hospitalisation for atrial fibrillation [41]; (ii) screening for paroxysmal atrial fibrillation [42]; and (iii) detecting recurrent atrial fibrillation after ablation or cardioversion [43]. In each of these examples an accurate QRS detector is a key step in processing the intermittent ECGs acquired by patients at home, where signal quality may be lower than in the clinical setting.

### Conclusion

This study identified two leading QRS detectors for use with single-lead, telehealth ECGs: the *nk* and *unsw* algorithms. These algorithms provided accurate QRS detection with fast execution times. Whilst most other algorithms performed well on data collected under clinical supervision, many did not perform as well on telehealth data, demonstrating the importance of selecting a high-performance algorithm for use in clinical analysis. The performance of even the leading algorithms was substantially lower on low-quality telehealth ECGs, highlighting the need to handle low-quality ECGs appropriately in an analysis pipeline. All the QRS detection algorithms used in this study are openly available, ensuring that they can be quickly used in future research. Furthermore, the code used to assess algorithm performance is also available to facilitate future research, at: https://github.com/floriankri/ecg_detector_assessment.

### Supporting information

**S1 Text. Supplementary Material.** The Supplementary Material provides additional results, details of the study methodology, and links to algorithms and datasets.
(PDF)

### Acknowledgments

[5] provided the foundations for the selection of datasets and their presentation in Table 2. ChatGPT (OpenAI, San Francisco, CA, USA) was used for language editing.

The views expressed are those of the author(s) and not necessarily those of the NIHR or the Department of Health and Social Care.

## Author Contributions

**Conceptualization:** Florian Kristof, Kate Williams, Peter H. Charlton.

**Data curation:** Florian Kristof, James Brimicombe, Andrew Dymond, Hannah Clair Lindén, Kate Williams, Peter H. Charlton.

**Formal analysis:** Florian Kristof, Leon Nissen, Peter H. Charlton.

**Funding acquisition:** Martin R. Cowie, Gregory Y. H. Lip, Jonathan Mant, Peter H. Charlton.

**Investigation:** Florian Kristof, James Brimicombe, Martin R. Cowie, Andrew Dymond, Hannah Clair Lindén, Gregory Y. H. Lip, Kate Williams, Jonathan Mant, Peter H. Charlton.

**Methodology:** Florian Kristof, Maximilian Kapsecker, Leon Nissen, James Brimicombe, Andrew Dymond, Hannah Clair Lindén, Gregory Y. H. Lip, Kate Williams, Jonathan Mant, Peter H. Charlton.

**Project administration:** Florian Kristof, Kate Williams, Jonathan Mant, Peter H. Charlton.

**Software:** Florian Kristof, Maximilian Kapsecker, Leon Nissen, Zixuan Ding, Peter H. Charlton.

**Supervision:** Jonathan Mant, Peter H. Charlton.

**Validation:** Florian Kristof.

**Writing – original draft:** Florian Kristof, Maximilian Kapsecker, Leon Nissen, Peter H. Charlton.

**Writing – review & editing:** Florian Kristof, Maximilian Kapsecker, Leon Nissen, James Brimicombe, Martin R. Cowie, Zixuan Ding, Andrew Dymond, Stephan M. Jonas, Hannah Clair Lindén, Gregory Y. H. Lip, Kate Williams, Jonathan Mant, Peter H. Charlton.

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
