## [Decision Letter · Decision Letter 0]

20 Feb 2024

PDIG-D-24-00016

QRS detection in single-lead, telehealth electrocardiogram signals: benchmarking open-source algorithms

PLOS Digital Health

Dear Dr. Charlton,

Thank you for submitting your manuscript to PLOS Digital Health. After careful consideration, we feel that it has merit but does not fully meet PLOS Digital Health's publication criteria as it currently stands. Therefore, we invite you to submit a revised version of the manuscript that addresses the points raised during the review process.

Please submit your revised manuscript within 60 days Apr 20 2024 11:59PM. If you will need more time than this to complete your revisions, please reply to this message or contact the journal office at digitalhealth@plos.org. Please include the following items when submitting your revised manuscript:

We look forward to receiving your revised manuscript.

Kind regards,

Calvin Or, PhD

Section Editor

PLOS Digital Health

Journal Requirements:

1. Please provide separate figure files in .tif or .eps format only and remove any figures embedded in your manuscript file. Please also ensure that all files are under our size limit of 10MB.

2. We ask that a manuscript source file is provided at Revision. Please upload your manuscript file as a .doc, .docx, .rtf or .tex.

Additional Editor Comments (if provided):

Reviewers' comments:

Reviewer's Responses to Questions

**Comments to the Author**

1. Does this manuscript meet PLOS Digital Health’s publication criteria? Is the manuscript technically sound, and do the data support the conclusions? The manuscript must describe methodologically and ethically rigorous research with conclusions that are appropriately drawn based on the data presented.

Reviewer #1: Yes

Reviewer #2: Yes

Reviewer #3: Partly

2. Has the statistical analysis been performed appropriately and rigorously?

Reviewer #1: Yes

Reviewer #2: Yes

Reviewer #3: Yes

3. Have the authors made all data underlying the findings in their manuscript fully available (please refer to the Data Availability Statement at the start of the manuscript PDF file)?

Reviewer #1: Yes

Reviewer #2: Yes

Reviewer #3: Yes

4. Is the manuscript presented in an intelligible fashion and written in standard English?

Reviewer #1: Yes

Reviewer #2: Yes

Reviewer #3: Yes

5. Review Comments to the Author

Reviewer #1: This study undertakes an extensive statistical analysis, amalgamating QRS detectors with diverse datasets and employing F1 score and error rate as performance metrics. However, a meticulous examination of the congruence between reported results in Figure 3 and Figure 4 is recommended to ensure coherence. Specifically, the instance where both the F1 score and error rate for the cell (mart, SAFER-nonAF-LOW) are recorded as 0 necessitates careful scrutiny, as their concurrent occurrence may signal potential calculation errors.

Furthermore, the application of the Mann-Whitney U test to assess performance across various subsets is acknowledged. Nonetheless, the absence of a comprehensive presentation of detailed testing results and statistical values introduces a notable limitation. To enhance the study's robustness, it is advisable to furnish explicit p-values and statistics derived from the Mann-Whitney U test. Relying solely on boxplot representations poses a potential constraint, particularly when disparities between median values seemingly contradict the non-significant (ns) result, assumed to arise from the U test. A more exhaustive reporting of statistical outcomes is imperative for a nuanced and compelling interpretation of the comparative analyses.

Reviewer #2: This paper benchmarked 18 open-source QRS detection algorithms to identify the best-performing one for ECG signals. The study compares their performance across datasets, including a novel dataset collected during AF screening. The manuscript has good quality and the results are interesting. I have a few comments:

1. Figure 7,8,9, without any explanation. What is the purpose of doing these experiments, what message the results convey?

2. What is the computing environment of running python and matlab program? How many CPU cores are used? Are there any parallelization used?

3. Could the authors specify what are the number of male/female samples in each dataset?

Reviewer #3: This work describes the performance of several QRS detection algorithms. While the work is not novel, in and of itself, there certainly is a welcome place in the literature for a thorough evaluation of these approaches. As the authors' note, QRS detection forms an important part of several clinically relevant tasks (e.g., arrhythmia detection, HRV analyses, etc.). However, the work suffers from technical flaws that require clarification before it can be published. 

1) The authors definitions of sensitivity and specificity are not correct as written and it is not clear whether this represents a typographical error or a true error in how these values were calculated. For example, the authors suggest that the numerator for both sensitivity calculations and for positive predictive value calculations are the same. This is incorrect. The sensitivity is the true positive rate and therefore the numerator refers to the number of reference QRS annotations that are also correct according to the algorithm. By contrast, numerator in the equation for the positive predictive value (PPV) is the number of algorithmic predictions that are correct. This is a very important point that needs to be clarified/corrected.

2) The authors define a correct prediction as one that is within +/- 150ms of the reference QRS annotation. This is a large range. In the introduction, the authors point to HRV and arrhythmia detection as important tasks that depend on QRS detection. However, such a large range (+/- 150ms of the QRS reference) will certainly not lead to accurate HRV estimates and will certainly not help with arrhythmia detection. The fact that this has been used in other studies is a poor reason to reply on this standard here. It is imperative that the authors discuss results using cutoffs that will yield more reliable HRV estimates (e.g., +/- 40ms).

6. PLOS authors have the option to publish the peer review history of their article (what does this mean?). If published, this will include your full peer review and any attached files.

**Do you want your identity to be public for this peer review?** For information about this choice, including consent withdrawal, please see our Privacy Policy.

Reviewer #1: No

Reviewer #2: No

Reviewer #3: No

---

## [Editor Report · Decision Letter 1]

27 May 2024

QRS detection in single-lead, telehealth electrocardiogram signals: benchmarking open-source algorithms

PDIG-D-24-00016R1

Dear Dr Charlton,

We are pleased to inform you that your manuscript 'QRS detection in single-lead, telehealth electrocardiogram signals: benchmarking open-source algorithms' has been provisionally accepted for publication in PLOS Digital Health.

Best regards,

Calvin Or, PhD

Section Editor

PLOS Digital Health

Reviewer Comments (if any, and for reference):

Reviewer's Responses to Questions

**Comments to the Author**

1. If the authors have adequately addressed your comments raised in a previous round of review and you feel that this manuscript is now acceptable for publication, you may indicate that here to bypass the “Comments to the Author” section, enter your conflict of interest statement in the “Confidential to Editor” section, and submit your "Accept" recommendation.

Reviewer #2: All comments have been addressed

2. Does this manuscript meet PLOS Digital Health’s publication criteria? Is the manuscript technically sound, and do the data support the conclusions? The manuscript must describe methodologically and ethically rigorous research with conclusions that are appropriately drawn based on the data presented.

Reviewer #2: Yes

3. Has the statistical analysis been performed appropriately and rigorously?

Reviewer #2: N/A

4. Have the authors made all data underlying the findings in their manuscript fully available (please refer to the Data Availability Statement at the start of the manuscript PDF file)?

Reviewer #2: No

5. Is the manuscript presented in an intelligible fashion and written in standard English?

Reviewer #2: (No Response)

6. Review Comments to the Author

Reviewer #2: "To address this, we have provided a repository of open-source algorithms and assessment code to accompany this article: https://github.com/floriankri/ecg_detector_assessment".

The link does not exist. If the authors do not plan to make code available, I suggest remove the false statement.

7. PLOS authors have the option to publish the peer review history of their article (what does this mean?). If published, this will include your full peer review and any attached files.

**Do you want your identity to be public for this peer review?** For information about this choice, including consent withdrawal, please see our Privacy Policy.

Reviewer #2: No
